# Unilateral Cauda Equina Syndrome Due to Cancer Metastasis Diagnosed with Electromyography: A Case Report

**DOI:** 10.3390/healthcare9101370

**Published:** 2021-10-14

**Authors:** Chan-Hee Park, Eunhee Park, Tae-Du Jung

**Affiliations:** 1Department of Rehabilitation Medicine, Kyungpook National University Hospital, Daegu 41944, Korea; chany9090@gmail.com; 2Department of Rehabilitation Medicine, Kyungpook National University Chilgok Hospital, Daegu 41404, Korea; ehmdpark@naver.com; 3Department of Rehabilitation Medicine, School of Medicine, Kyungpook National University, Daegu 41944, Korea

**Keywords:** cauda equina syndrome, unilateral cauda equina syndrome, metastasis, electromyography

## Abstract

*Background*: Typical cauda equina syndrome (CES) presents as low back pain, bilateral leg pain with motor and sensory deficits, genitourinary dysfunction, saddle anesthesia and fecal incontinence. In addition, it is a neurosurgical emergency, which is essential to diagnose as soon as possible, and needs prompt intervention. However, unilateral CES is rare. Here, we report a unique case of a patient who had unilateral symptoms of CES due to cancer metastasis and was diagnosed through electromyography. *Methods*: A 71-year-old man with diffuse large B cell lymphoma (DLBCL) suffered from severe pain, motor weakness in the right lower limb and urinary incontinence, and hemi-saddle anesthesia. It was easy to be confused with lumbar radiculopathy due to the unilateral symptoms. Lumbar spine magnetic resonance imaging (MRI) showed suspected multifocal bone metastasis in the TL spine, including T11-L5, the bilateral sacrum and iliac bones, and suspected epidural metastasis at L4/5, L5/S1 and the sacrum. PET CT conducted after the third R-CHOP showed residual hypermetabolic lesions in L5, the sacrum, and the right presacral area. *Results*: Nerve conduction studies (NCS) revealed peripheral neuropathy in both hands and feet. Electromyography (EMG) presented abnormal results indicating development of muscle membrane instability following neural injury, not only on the right symptomatic side, but also on the other side which was considered intact. Overall, he was diagnosed with cauda equina syndrome caused by DLBCL metastasis, and referred to neurosurgical department. *Conclusions*: Early diagnosis of unilateral CES may go unnoticed due to its unilateral symptoms. Failure to perform the intervention at the proper time can impede recovery and leave permanent complications. Therefore, physicians need to know not only the typical CES, but also the clinical features of atypical CES when encountering a patient, and further evaluation such as electrodiagnostic study or lumbar spine MRI have to be considered.

## 1. Introduction

Cauda equina syndrome (CES) refers to a constellation of signs and symptoms that result from damage to the cauda equina, which refers to the portion of the nervous system below the conus medullaris and consists of peripheral nerves, both motor and sensory, within the spinal canal and thecal sac [1]. The typical symptoms include low back pain, bilateral leg pain with motor and sensory deficits, genitourinary dysfunction, saddle anesthesia and fecal incontinence [2]. CES is a neurosurgical emergency, unlike typical symptomatic herniated lumbar intervertebral discs. Therefore, early diagnosis and prompt surgical intervention is essential [3]. Unlike typical CES, some patients have those symptoms in only the unilateral side, unilateral sensibility loss, weakness, and hemi-saddle anesthesia. It was defined as hemi-cauda equina syndrome and was treated as a neurosurgical emergency, like CES [4]. Unilateral cauda equina syndrome, which requires emergency surgical treatment, can be easily mistaken for simple unilateral radiculopathy due to its unilateral symptoms. Therefore, through this case, which was successful in diagnosis through electromyography (EMG), we report to enable appropriate early treatment without missing the critical time that can leave lifelong sequelae due to late diagnosis or misdiagnosis. 

## 2. Case Report

A 71-year-old male, a worker at factory processing agricultural products, had suffered from right whole leg pain for five months. One month later, weakness of the right leg and ankle developed. He was diagnosed with hypertension, benign prostate hyperplasia (BPH), and diffuse large B cell lymphoma (DLBCL) through neck lymph node biopsy. Then, he underwent a rituximab, cyclophosphamide, hydroxydaunorubicin, oncovin, and prednisone (R-CHOP) regimen of chemotherapy. Despite taking painkillers, his right leg pain and weakness persisted without improvement. At first, the Hematology and Oncology Department suspected unilateral radiculopathy from the patient's symptoms, and five months after the initial onset, he was referred to the Rehabilitation Department.

On physical examination, according to the medical research council (MRC) grading system, the motor strength of right hip flexor, knee extensor, ankle dorsiflexor and ankle plantar flexor was 2, 2, 0, and 0, respectively. There were no subjective complaints about the patient's left lower limb. Both upper limbs and trunk power was normal. There were decreased sensations in the whole right leg from L1 to S4-5 dermatomes, and he had saddle anesthesia only on the right side. Knee jerk and ankle jerk was decreased in both sides. Through more detailed investigation of the patient’s history before electrodiagnostic examination, it was revealed that the patient had urinary incontinence five months ago without fecal incontinence, and he had a surgery for BPH. After surgery, urinary incontinence was slightly improved, but nevertheless persisted. He also had had tingling sensations without weakness in both hands and feet since several years ago. 

At the time of admission to the Hematology Oncology Department, an MRI of the lumbar spine showed suspicious findings of multifocal bone metastasis in the thoracic-lumbar spine, including T11-L5 and bilateral sacrum, iliac bones and epidural metastasis at L4/5, L5/S1 and the sacrum (Figure 1). PET-CT was also performed after the third R-CHOP at the time of admission to the Department of Hematology and Oncology, and showed residual hypermetabolic lesions in L5, the sacrum, and the right presacral area (Figure 2). Electrodiagnostic examination was performed five months after the onset (Table 1).

Before EMG, a nerve conduction study (NCS) of the lower extremities was first performed. In the motor conduction study, extensor digitorum brevis (EDB), abductor hallucis (AH), and rectus femoris (RF) were performed for the bilateral common peroneal nerve, tibial nerve, and femoral nerve, respectively. The test was performed by recording from those muscles. In the sensory conduction study, bilateral superficial peroneal nerve, sural nerve, and saphenous nerve were tested. As will be explained in detail later, it was confirmed that the sensory nerves of both lower extremities were degraded on the NCS. Therefore, to differentiate the possibility of overlapping peripheral neuropathy, NCS of both upper extremities was also performed. In the case of both upper extremities, conduction studies were performed on the motor and sensory nerves for the median, ulnar, and radial nerves, respectively.

In the motor nerve conduction study, Rt. common peroneal and tibial nerves showed no response. The amplitude of Lt. abductor hallucis and bilateral rectus femoris muscle recorded compound motor unit action potential (CMAP) was decreased. Distal latency and conduction velocity were all within the normal range. In the sensory, there was no response to Rt. saphenous, superficial peroneal, and sural nerve stimulation. On the left side, amplitude reduction was observed in all tested lower limb nerves. Peak latency was in the normal range. In the upper extremities, CMAP and sensory nerve action potential (SNAP) amplitudes were reduced in both median and ulnar nerves.

Needle electromyography showed unexpected abnormal results not only in the right symptomatic side, but also in the other side which was considered not involved. There were abnormal spontaneous activities in most of the bilateral lower extremities except the hip flexor and knee extensor muscles. Fibrillation potentials and positive sharp waves were noted in bilateral paraspinalis muscles and the external sphincter (Table 2). Those abnormal EMG findings indicated that the development of muscle membrane instability following neural injury was prominent on the right side. Pudendal somatosensory evoked potential (SEP), and bulbocavernosus reflex were not performed. Cauda equina syndrome is usually bilateral, and the abnormal pattern is observed on both sides in motor conduction studies and EMG at multiple levels below the lesion, including the external anal sphincter [5]. Although there were only unilateral symptoms, it was considered to be consistent with the NCS and EMG test findings of cauda equina syndrome, in that an abnormal pattern was confirmed bilaterally and multilevel on the test results. Overall, electrodiagnostic results, along with the patient’s clinical presentation and MRI findings, led us to a diagnosis of cauda equina syndrome with peripheral neuropathy.

The patient was referred to neurosurgery, and then after consultation with the hemato-oncologist about his life expectancy and risk-benefit, he decided to maintain conservative treatment including R-CHOP for DLBL, rather than a surgical approach for CES.

## 3. Discussion

The most common cause for CES is lumbar disc prolapse, though it can also be caused by trauma, spinal canal stenosis, tumors, epidural hematoma or abscess, inferior vena cava thrombosis and spinal manipulation [6,7]. The most important thing in CES management is early diagnosis, and ensuring rapid transfer to hospital [2]. There is still no consensus on the urgency of surgical decompression, but many authors still recommend surgery as soon as possible to maximize functional recovery, and features of CES such as saddle anesthesia, and bladder and bowel dysfunction are considered as ‘red flag’ symptoms [2,3]. The classical presentation of CES includes low back pain, bilateral sciatica with motor and sensory deficits, genitourinary dysfunction, saddle anesthesia and fecal incontinence [6]. However, several case studies have been reported with unilateral symptoms, not bilateral sides [4].

Our patient had weakness of lower leg only on the right side, urinary incontinence and hemi-saddle anesthesia, with MRI finding suspected epidural metastasis at L4/5, L5/S1 and the sacrum. In the case of unilateral cauda equina, the fact that it is unilateral is easy to confuse with unilateral radiculopathy, and in particular, if there is an underlying urogenital disease such as BPH, it is more likely to be masked, so it is necessary to be careful in treating patients like our case. Diagnosis with cauda equina syndrome should not be missed because it is a neurological emergency. When unilateral symptoms of CES are suspected, it is necessary to perform electrodiagnostic evaluation or L-spine MRI to distinguish unilateral cauda equina syndrome.

We only performed the NCS and EMG tests of the upper and lower extremities, but the bulbocavernosus reflex test and pudendal SEP, which would be a "no response" if there was an abnormality, were not additionally performed because the patient was having a hard time due to the long-term examination. As our patient has an underlying disease such as BPH, it would be helpful for diagnosis if the above test is additionally performed to determine whether the cause of urinary symptoms is due to an underlying disease or whether there is actually damage to the nerve.

In our opinion, unlike cauda equina syndrome, which is usually caused by disc prolapse, if the etiology of the patient has a possibility of diffuse involvement, such as epidural metastasis, as in our case, the hidden cauda equina syndrome is unilateral even if the symptoms are unilateral due to the nature of the metastasis. It is thought that confirmation through nerve conduction and electromyography examination is necessary.

Although we report a single case, we think that additional research is needed to determine whether there are any other influencing factors and their relevance through a large-scale multi-center.

This atypical case of CES is rare, and to our knowledge, the EMG results of hemi-cauda equina have not been reported yet. The patient only had unilateral symptoms, and has been greatly assisted in the diagnosis of cauda equina syndrome by conducting electromyography. It is clinically significant in that we report needle EMG, as well as the nerve conduction study result, of a patient with unilateral CES, and show the importance of early diagnosis through detailed examination and electrodiagnostic study, even if unilateral symptoms are presented.

## 4. Conclusions

In conclusion, it is important not to miss the critical time in the treatment of unilateral cauda equina syndrome, which can be easily mistaken for simple unilateral radiculopathy. In this case, in addition to the neurologic department, various departments including cancer-related departments should conduct nerve conduction and EMG tests, in addition to imaging tests such as spine MRI, when confronted with such patients, or cooperate with related departments to enable early detection and treatment.

## Figures and Tables

**Figure 1 healthcare-09-01370-f001:**
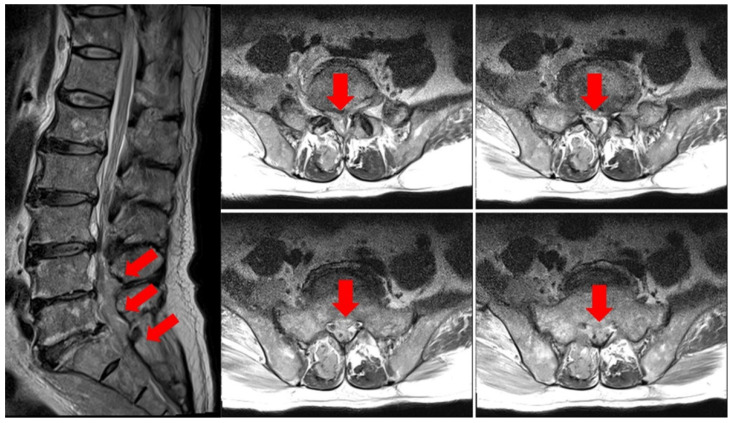
Lumbar spine MRI T2 weighted image. Epidural metastasis at L4/5, L5/S1, and the sacrum is noted (arrows).

**Figure 2 healthcare-09-01370-f002:**
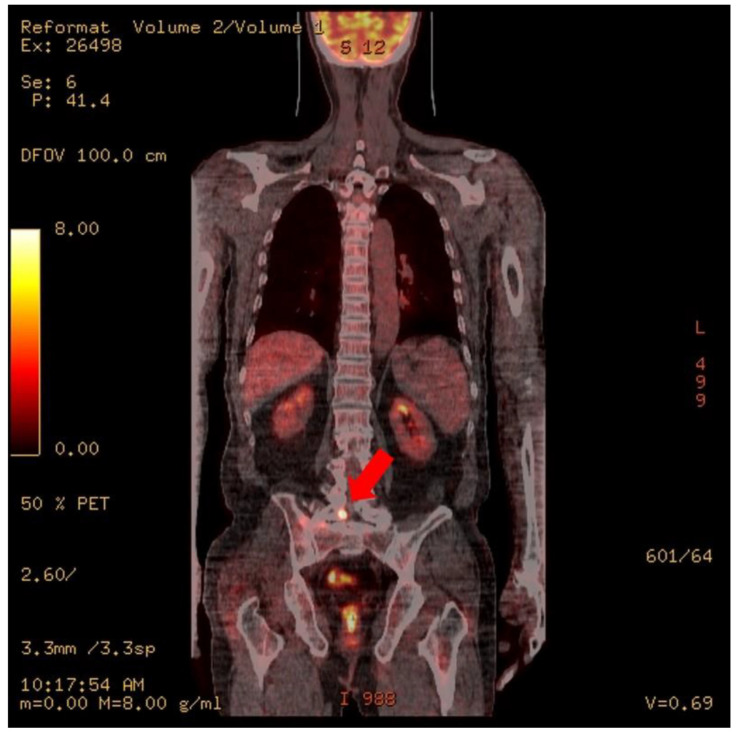
PET CT. Focal increased uptake is noted at right L5, the sacrum, and the presacral area (arrows).

**Table 1 healthcare-09-01370-t001:** Result of nerve conduction study.

Motor Nerve Conduction					Sensory Nerve Conduction			
Side	Nerve	Latency (msec)	Amplitude (mV)	CV (m/sec)	Side	Nerve	Peak latency (msec)	Amplitude (mV)
Right	Median (APB)	4.01	5.6	56.2	Right	Median	4.17	8.40
	Ulnar (ADM)	3.13	7.6	63.2		Ulnar	NR	NR
	Femoral (RF)	6.46	0.3			Saphenous	NR	NR
	Common peroneal (EDB)	NR	NR	NR		Superficial peroneal	NR	NR
	Tibial (AH)	NR	NR	NR		Sural	NR	NR
Left	Median (APB)	3.8	6.3	57.6	Left	Median	4.11	8.50
	Ulnar (ADM)	2.76	7.7	63.2		Ulnar	NR	NR
	Femoral (RF)	5.83	1.4			Saphenous	2.76	2.60
	Common peroneal (EDB)	4.64	1.7	34.7		Superficial peroneal	2.50	2.70
	Tibial (AH)	5.26	1.2	51.2		Sural	2.50	4.00

CV, conduction velocity; APB, abductor pollicis brevis; ADM, abductor digiti minimi; RF, rectus femoris; EDB, extensor digitorum brevis; AH, abductor hallucis; NR, no response.

**Table 2 healthcare-09-01370-t002:** Result of electromyography.

Side	Muscle	ASA		Motor Unit Potentials			Interference Pattern
		Fibs	PSW	Polyphasic	Amplitude	Duration	
Right	Iliopsoas	-	-	-	-	-	Discrete
	Vastus medialis	-	-	Increased	-	-	Reduced
	Tibialis anterior	3+	3+	-	-	-	Zero
	Peroneus longus	1+	-	-	-	-	Zero
	Tensor fascia lata	-	-	-	-	-	Zero
	Gastrocnemius	-	3+	-	-	-	Zero
	Biceps femoris (SH)	2+	2+	-	-	-	Zero
	Gluteus maximus	-	2+	Increased	-	Long	Single
	External anal sphincter	1+	1+	-	-	-	Zero
	L5 paraspinalis	2+	2+				
Left	Iliopsoas	-	-	-	-	-	Complete
	Vastus medialis	-	-	-	-	-	Reduced
	Tibialis anterior	-	1+	-	High	-	Reduced
	Gastrocnemius	1+	-	Increased	Giant	-	Reduced
	Biceps femoris (SH)	1+	1+	Increased	-	-	Reduced
	Gluteus maximus	-	-	-	-	-	Complete
	External anal sphincter	1+	1+	-	-	-	Zero
	L5 paraspinalis	2+	1+				

ASA, abnormal spontaneous activities; Fibs, fibrillation potentials; PSW, positive sharp waves; SH, short head; L, lumbar.

## Data Availability

Available upon reasonable request.

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
