# Peer review of "Unilateral Cauda Equina Syndrome Due to Cancer Metastasis Diagnosed with Electromyography: A Case Report"

_healthcare, 2021, doi:10.3390/healthcare9101370_

Round 1
Reviewer 1 Report
This case report is well written and well described, so this referee is satisfied.
Author Response
Thank you for your kind review of my case report.
Reviewer 2 Report
- The authors do not define the objective of the work. They only report that they "will describe unilateral CES diagnosed with electromyography (EMG) which is easy to be confused with radiculopathy".
-
What is this information in the middle of the article (end of page 4 and the beggining of page 5) ????
""Materials and Methods should be described with sufficient details to allow others to replicate and build on the published results. Please note that the publication of your man-uscript implicates that you must make all materials, data, computer code, and protocols
associated with the publication available to readers. Please disclose at the submission stage any restrictions on the availability of materials or information. New methods and protocols should be described in detail while well-established methods can be briefly de-scribed and appropriately cited. "" - Methods and discussion lack depth and granularity. And there is no discussion about EMG signals for CES.
- Given that this is a case study, the conclusion is fragile.
Reviewer 3 Report
Thank you for an interesting case report.
Author Response

(The authors gave the same response as above.)

Reviewer 4 Report
This is an interesting case report
Author Response

(The authors gave the same response as above.)
